# Role of Dorsomedial Hypothalamus GABAergic Neurons in Sleep–Wake States in Response to Changes in Ambient Temperature in Mice

**DOI:** 10.3390/ijms23031270

**Published:** 2022-01-23

**Authors:** Lei Li, Meng-Qi Zhang, Xiao Sun, Wen-Ying Liu, Zhi-Li Huang, Yi-Qun Wang

**Affiliations:** Department of Pharmacology, School of Basic Medical Sciences and State Key Laboratory of Medical Neurobiology and MOE Frontiers Center for Brain Science, Institutes of Brain Science, Fudan University, Shanghai 200032, China; 19111010089@fudan.edu.cn (L.L.); 14111010003@fudan.edu.cn (M.-Q.Z.); 18211010070@fudan.edu.cn (X.S.); 15111520011@fudan.edu.cn (W.-Y.L.)

**Keywords:** cold exposure, dorsomedial hypothalamus, GABAergic neurons, sleep–wake behaviors, warm exposure

## Abstract

Good sleep quality is essential for maintaining the body’s attention during wakefulness, which is easily affected by external factors such as an ambient temperature. However, the mechanism by which an ambient temperature influences sleep–wake behaviors remains unclear. The dorsomedial hypothalamus (DMH) has been reported to be involved in thermoregulation. It also receives projection from the preoptic area, which is an important region for sleep and energy homeostasis and the suprachiasmatic nucleus—a main control area of the clock rhythm. Therefore, we hypothesized that the DMH plays an important role in the regulation of sleep related to ambient temperatures. In this study, we found that cold exposure (24/20/16/12 °C) increased wakefulness and decreased non–rapid eye movement (NREM) sleep, while warm exposure (32/36/40/44 °C) increased NREM sleep and decreased wakefulness compared to 28 °C conditions in wild-type mice. Then, using non-specific and specific apoptosis, we found that lesions of whole DMH neurons and DMH γ–aminobutyric acid (GABA)-ergic neurons induced by caspase-3 virus aggravated the fluctuation of core body temperature after warm exposure and attenuated the change in sleep–wake behaviors during cold and warm exposure. However, chemogenetic activation or inhibition of DMH GABAergic neurons did not affect the sleep–wake cycle. Collectively, our findings reveal an essential role of DMH GABAergic neurons in the regulation of sleep–wake behaviors elicited by a change in ambient temperature.

## 1. Introduction

Sleep, core body temperature, and ambient temperature are tightly interconnected in homeothermic mammals [1]. Under physiological conditions, homeothermic animals have a complex central nervous system used for precise body temperature regulation [1,2]. Warm exposure is reported to promote non-rapid eye movement (NREM) sleep, rapid eye movement (REM) sleep, and recuperative sleep after sleep deprivation and to reduce the latency to sleep. By contrast, cold exposure induces arousal [3]. However, how the temperature signal interacts with sleep–wake behaviors and which brain regions are involved in this interaction remains unclear. 

The dorsomedial hypothalamus (DMH) is a brain region adjacent to the third ventricle and located caudal and ventral to the paraventricular nucleus of the hypothalamus, which is critical for arousal promotion and maintenance [4], food intake, and locomotor activity [5]. The lateral portion of the DMH is adjacent to both the fornix and the lateral hypothalamic area [6], which is a heterogeneous nucleus [7] associated with sleep–wake regulation [8], feeding behavior, and energy metabolism [9]. The DMH also receives projections from the center of body temperature regulation—the preoptic area (POA) [10] of the hypothalamus, and has bidirectional projections with the site of the principal circadian clock—the suprachiasmatic nucleus (SCN) [11,12,13,14,15,16]. The DMH is also involved in many other life activities, including feeding [17,18], emotional stress [19], locomotion activity [18], and sleep–wake behaviors [18,20,21,22]. Using micro-endoscopic calcium imaging and bidirectional optogenetic manipulation, Dan et al. found two distinct DMH galaninergic populations that separately projected to the POA and the raphe pallidus, suppressing and promoting REM sleep, respectively [23]. It has also been reported that both glutamatergic and γ–aminobutyric acid (GABA)-ergic neurons in the dorsal part of the dorsomedial hypothalamus (DMD) respond to changes in ambient temperatures, and the response of GABAergic neurons lasts longer than that of glutamatergic neurons [24]. Therefore, we hypothesized that the DMH GABAergic neurons, which are most densely distributed in the DMH, are involved in body temperature regulation during sleep in mice.

In this study, non-specific and specific apoptosis were employed to assess the necessity of the DMH neurons in the regulation of the sleep–wake cycle elicited by the change in ambient temperature. We used pharmacogenetic designer receptors exclusively activated by designer drugs (DREADD) and pharmacological approaches to manipulate neuronal activity to investigate whether the DMH plays a role in the regulation of sleep and homeostasis. Our results provide evidence of the role of DMH GABAergic neurons in the regulation of the sleep–wake cycle elicited by changes in ambient temperatures.

## 2. Results

### 2.1. Cold Exposure Increased Wakefulness While Warm Exposure Increased NREM Sleep in WT Mice

To determine the reliability of the method and explore the cold and warm exposure temperature suitable for this research, we studied the effects of gradient temperature exposure on sleep–wake behavior in wild type (WT) mice. According to previous research, ambient temperatures exert a prominent influence on sleep. In general, low ambient temperatures impair sleep, whereas higher temperatures tend to promote sleep [21,25]. According to the previous study, warm exposure (30 ± 1 °C) induced the significant increase of the total sleep time, light slow-wave sleep, and REM sleep compared to the baseline condition (24 ± 1 °C), especially during the dark period. On the contrary, slow-wave sleep and REM sleep were decreased after cold exposure (18 ± 1 °C) during both light and dark periods [3]. In our study, we performed cold exposure during the light phase while performing warm exposure during the dark phase. Cold exposure induced a more obvious waking-promoting effect when the animals were most sleepy during the light period (07:00–19:00), especially in 09:00–11:00. During the dark period (19:00–07:00), warm exposure at this stage (21:00–23:00) played a more significant role in promoting sleep when the animals were more active. Previous studies have shown that the neutral temperature range for rats and mice is between 25 °C and 32 °C [26]. Therefore, 28 °C was selected as the base temperature for this study.

To investigate the effects of gradient cold exposure on sleep–wake behavior in WT mice, we analyzed electroencephalogram (EEG) and electromyogram (EMG) recordings conducted at a constant temperature (28 ± 0.5 °C) for 48 h. On the second day, gradient cold exposure (24/20/16/12 °C) was given between 9:00 and 11:00, and the rest of the recording conditions were the same as on the first day (Figure 1A). Compared to the baseline recording on the first day, the 24/20/16/12 °C cold exposure increased wakefulness and decreased NREM sleep in the first hour (24 °C vs. 28 °C, *n* = 7, *p* < 0.05; 20/16/12 °C vs. 28 °C, *n* = 6–7, *p* < 0.01) (Figure 1B,C). When exposed to cold exposure at 16/12 °C, the amount of REM sleep in mice decreased during the first hour (*n* = 7, *p* < 0.05) (Figure 1D). During cold exposure (9:00–11:00), the total amount of wakefulness increased by 86.00%, 93.84%, 119.13%, or 95.91%, with a 34.18%, 53.61%, 49.38%, or 36.63% decrease in NREM sleep and a 57.23%, 53.52%, 62.72%, or 75.54% decrease in REM sleep elicited by 24 °C, 20 °C, 16 °C, or 12 °C cold exposure, respectively (vs. 28 °C, *n* = 6–7, *p* < 0.01). During cold exposure at 24 °C, 20 °C, and 16 °C, the increase in wakefulness gradually increased, while at 12 °C, wakefulness decreased slightly compared with that at 16 °C (Figure 1E–G). It is possible that the temperature was too low to cause hypothermia in mice, which, in turn, affected the response of the mice to the temperatures. To explore the detailed changes in sleep–wake cycle caused by cold exposure, we analyzed the sleep–wake architecture and sleep depth during 2 h of exposure to 20 °C. Cold exposure decreased the numbers of bouts of wakefulness (Figure 1H), REM sleep (Figure 1I), NREM sleep (Figure 1J), and the episode number of wakefulness (Figure 1M) but increased the mean duration of wakefulness significantly (Figure 1N). Cold exposure did not affect the sleep depth of NREM sleep compared to the baseline group (Figure 1K). The results indicate that cold exposure can increase wakefulness and reduce NREM sleep and REM sleep in mice.

To study the effects of gradient warm exposure on sleep–wake behavior in WT mice, we analyzed EEG and EMG recordings made at a constant temperature (28 ± 0.5 °C) for 48 h. On the second day, a gradient warm exposure (32/36/40/44 °C) was constructed between 21:00 and 23:00, and the rest of the recording conditions were the same as on the first day (Figure 2A). Compared to the baseline recording on the first day, the 32/36/40/44 °C warm exposure increased NREM sleep and decreased wakefulness during 21:00–23:00 (21:00: 36/40 °C vs. 28 °C, *p* < 0.05, *n* = 5/11; 44 °C vs. 28 °C, *p* < 0.01, *n* = 7; 22:00: 36/40/44 °C vs. 28 °C, *p* < 0.01, *n* = 5/7/11) (Figure 2B,C). During warm exposure (21:00–23:00), the total amount of NREM sleep increased by 49.09%, 256.20%, 435.15%, or 461.47%, with a 5.02%, 24.60%, 42.05%, or 44.58% decrease in wakefulness elicited by 32 °C, 36 °C, 40 °C, or 44 °C warm exposure, respectively (36/40/44 °C vs. 28 °C, *p* < 0.01, *n* = 5–13) (Figure 2E–G). During warm exposure at 36 °C, 40 °C, and 44 °C, the decrease in wakefulness gradually increased. To explore the detailed changes in sleep–wake cycle caused by warm exposure, we analyzed the sleep–wake architecture and sleep depth during 2 h of exposure to 36 °C. Warm exposure increased the numbers of bouts of wakefulness (Figure 2H), NREM sleep (Figure 2I), the transition number between NREM sleep and wakefulness (Figure 2J), and the episode number of wakefulness (Figure 2L) but decreased the mean duration of wakefulness and NREM sleep significantly, which demonstrated the fragmentation of wakefulness during warm exposure (Figure 2M). Warm exposure elicited the change in the frequency range of 0–1.0 Hz and 7–7.5 Hz of NREM sleep but did not affect the delta power of NREM sleep (Figure 2K). The results indicate that warm exposure can increase NREM sleep and reduce wakefulness in mice.

Based on the above results, 20 °C and 36 °C were selected as the cold and warm exposure temperatures in subsequent studies.

### 2.2. Whole DMH Neurons Are Necessary to Respond to Changes in Ambient Temperature during Sleep–Wake Behavior in Mice

Next, we ablated neurons within the DMH to test whether they are necessary for the response to cold/warm exposure during sleep–wake behavior and the core body temperature. To kill the neurons in the DMH, we bilaterally microinjected the mixture of an adeno-associated virus (AAV) vector that Cre-dependently expresses caspase-3 with a linker replaced with a Tobacco Etch Virus protease cleavage site (AAV–DIO–taCasp3–TEVp), an AAV vector that Cre-dependently expresses enhanced green fluorescent protein (AAV–EF1a–DIO–eGFP), and an AAV vector that expresses the Cre recombinase (AAV–hSyn–Cre) into the DMH of WT mice (DMH–WT–Casp3 mice). The WT mice injected with the mixture of AAV–hSyn–Cre, and EF1a–DIO–eGFP–AAV in the DMH (DMH–WT–eGFP mice) served as control by expressing eGFP in the DMH neurons (Figure 3A–C). The DMH neurons were efficiently eliminated by 46.88% in DMH–WT–Casp3 mice (Figure 3D). Three weeks after AAV injection, we performed the locomotion activity, core body temperature, and EEG and EMG recordings of DMH–WT–Casp3 mice and DMH–WT–eGFP mice. 

To investigate whether the whole DMH neurons are involved in the thermoregulatory response elicited by cold and warm exposure, we compared the changes in core body temperature between DMH–WT–Casp3 and DMH–WT–eGFP mice before, during, and after cold exposure. We analyzed the core body temperature obtained at a constant temperature (28 ± 0.5 °C) for 48 h. On the second day, 20 °C cold exposure or 36 °C warm exposure was given between 09:00 and 11:00 or between 21:00 and 23:00, and the rest of the recording conditions were the same as on the first day (Figure 3E and Figure 4A). The results show that neither the apoptosis of DMH neurons nor 20 °C cold exposure affected the core body temperature of mice (*n* = 11) (Figure 3F–I). In other words, when exposed to cold temperatures, mice can increase heat production and reduce heat dissipation through temperature regulation mechanisms to maintain a constant core body temperature. The DMH neurons are not necessary in this process, suggesting that the DMH might not be involved in body temperature regulation elicited by cold exposure. By contrast, compared with DMH–WT–eGFP mice, 36 °C warm exposure elicited a 2 h drop (23:00–01:00) in the core body temperature of DMH–WT–Casp3 mice, and the average body temperature during the active period after the end of warm exposure decreased by 0.88 °C (*n* = 8–10) (Figure 4B–E). After a 2 h (23:00–01:00) drop, it gradually returned to a normal level (*n* = 8, *p* < 0.05) (Figure 4C).

The results suggest that warm exposure may lead to increased heat production and reduced heat dissipation in mice. After the end of warm exposure, because the above-mentioned temperature regulation behavior was not completely corrected, it showed a delayed temperature drop. The whole DMH neurons play a role in promoting heat production during recovery from hypothermia. The process is disturbed in DMH–WT–Casp3 mice, so the core body temperature of the mice decreases more drastically after the end of warm exposure.

To investigate whether the whole DMH neurons are involved in the sleep–wake regulation elicited by cold/warm exposure, we analyzed the EEG and EMG performed at a constant temperature (28 ± 0.5 °C) for 48 h. On the second day, 20 °C cold exposure or 36 °C warm exposure was given between 09:00 and 11:00 or between 21:00 and 23:00, and the rest of the recording conditions were the same as on the first day (Figure 3E and Figure 4A). The results revealed that cold exposure significantly increased arousal time and reduced NREM sleep and REM sleep in DMH–WT–eGFP (*n* = 11, *p* < 0.05) and DMH–WT–Casp3 (*n* = 7, *p* < 0.05) mice during the first hour. However, compared with the baseline condition at 28 °C, during cold exposure, the amount of time spent in wakefulness increased by 27.06 min in the control group, and the time in NREM sleep decreased by 22.49 min in DMH–WT–eGFP mice (*n* = 11, *p* < 0.01). The amount of time spent in wakefulness increased by 15.15 min, and the time in NREM sleep decreased by 10.52 min (*n* = 7, *p* < 0.01) in DMH–WT–Casp3 mice. Compared to the control group, the amount of time DMH–WT–Casp3 mice spent in wakefulness during cold exposure decreased by 44.01% and the time in NREM sleep increased by 53.22% (Figure 3P,Q).

Warm exposure significantly increased NREM and REM sleep and reduced wakefulness in DMH–WT–eGFP (*n* = 9, *p* < 0.05) and DMH–WT–Casp3 (*n* = 6, *p* < 0.05) mice during the first hour. However, compared to the baseline condition at 28 °C, during warm exposure, the amount of time spent in NREM sleep increased by 27.39 min in the control group, and the time spent in wakefulness decreased by 27.72 min in DMH–WT–eGFP mice (*n* = 9, *p* < 0.01). Meanwhile, the amount of time spent in NREM sleep increased by 13.70 min, and the time in wakefulness decreased by 10.40 min (*n* = 6, *p* < 0.01) in DMH–WT–Casp3 mice. Compared with the control group, the amount of time spent in NREM sleep by DMH–WT–Casp3 mice during warm exposure decreased by 49.98%, and the time spent in wakefulness increased by 62.48% (Figure 4L,M).

The above results indicate that the non-selective apoptosis of the whole DMH neurons attenuates the wakefulness elicited by cold exposure and NREM sleep elicited by warm exposure.

### 2.3. Response to Changes in Ambient Temperature during Sleep–Wake Behavior in Mice Requires DMH GABAergic Neurons

To investigate whether GABAergic neurons in the DMH are involved in the response to changes in ambient temperature during sleep–wake behavior, we ablated GABAergic neurons specifically within the DMH in vesicular GABA transporter (vGAT) -Cre mice.

For lesioning of GABAergic neurons in the DMH, we bilaterally microinjected the mixture of an AAV vector that Cre-dependently expresses caspase-3 (AAV–DIO–taCasp3–TEVp) and an AAV vector that Cre-dependently expresses eGFP (AAV–EF1a–DIO–eGFP) into the DMH in vGAT–Cre mice. The vGAT–Cre mice injected with the AAV–EF1a–DIO–eGFP in the DMH (DMH–vGAT–eGFP mice) served as control by expressing eGFP in the DMH GABAergic neurons. The DMH GABAergic neurons were efficiently eliminated in DMH–vGAT–Casp3 mice (Figure 5A–C). The DMH GABAergic neurons were efficiently eliminated by 51.06% in DMH–vGAT–Casp3 mice (Figure 5D). Three weeks after AAV injection, we performed the locomotion activity, core body temperature, and EEG and EMG recordings of DMH–vGAT–Casp3 and DMH–vGAT–eGFP mice. Compared to the controls, the apoptosis of DMH GABAergic neurons did not affect circadian rhythms or overall levels of the core body temperature and locomotion under baseline conditions in mice.

To investigate whether the DMH GABAergic neurons are involved in the thermoregulatory response elicited by cold and warm exposure, we compared the changes in core body temperature between DMH–vGAT–Casp3 and DMH–vGAT–eGFP mice before and after cold exposure. We analyzed the core body temperature obtained at a constant temperature (28 ± 0.5 °C) for 48 h. On the second day, 20 °C cold exposure or 36 °C warm exposure was given between 09:00 and 11:00 or 21:00 and 23:00, and the rest of the recording conditions were the same as on the first day (Figure 5E and Figure 6A). The results show that 20 °C cold exposure does not affect the core body temperature of control mice (*n* = 6) or DMH–vGAT–Casp3 mice (*n* = 7) (Figure 5F–I). This suggests that the DMH might not be involved in body temperature regulation elicited by cold exposure, or the loss of DMH function caused by chronic GABAergic neurons apoptosis can be compensated by other body temperature regulating pathways.

By contrast, during 36 °C warm exposure, the core body temperature of the control mice did not change significantly (Figure 6B–E) (*n* = 6). In DMH–vGAT–Casp3 mice, the core body temperature increased by 0.28 °C (*n* = 7) in the first hour of warm exposure (21:00–22:00) and then decreased from the second hour. After four hours (23:00–3:00), it gradually returned to a normal level (*n* = 7, *p* < 0.01) (Figure 6C). These results suggest that GABAergic neurons in the DMH play an important role in thermogenesis during hypothermia recovery.

To investigate whether the DMH neurons are involved in the sleep–wake regulation elicited by cold/warm exposure, we analyzed the EEG and EMG performed at a constant temperature (28 ± 0.5 °C) for 48 h. On the second day, 20 °C cold exposure or 36 °C warm exposure was given between 09:00 and 11:00 or between 21:00 and 23:00, and the rest of the recording conditions were the same as on the first day (Figure 5E and Figure 6A). The results revealed that cold exposure significantly increased arousal and reduced NREM sleep and REM sleep in DMH–vGAT–eGFP and DMH–vGAT–Casp3 mice. However, compared to the baseline condition at 28 °C, during cold exposure, the amount of time spent in wakefulness increased by 26.14 min in the control group, and the time spent in NREM sleep decreased by 20.59 min in DMH–vGAT–eGFP mice (*n* = 6, *p* < 0.01). Meanwhile, the amount of time spent in wakefulness increased by 16.12 min, and the time in NREM sleep decreased by 12.71 min (*n* = 8, *p* < 0.01) in DMH–vGAT–Casp3 mice. Compared with the control group, the amount of time spent in wakefulness in DMH–vGAT–Casp3 mice during cold exposure decreased by 38.33%, and the time in NREM sleep increased by 38.27% (Figure 5P,Q).

Warm exposure significantly increased NREM and REM sleep and reduced wakefulness in DMH–vGAT–eGFP (*n* = 7, *p* < 0.05) and DMH–vGAT–Casp3 mice (*n* = 8, *p* < 0.05) during the first hour. However, compared with the baseline condition at 28 °C, during warm exposure, the amount of time spent in NREM sleep increased by 27.57 min in the control group, and the time spent in wakefulness decreased by 26.62 min in DMH–vGAT–eGFP mice (*n* = 7, *p* < 0.01). Meanwhile, the amount of time spent in NREM sleep increased by 6.70 min, and the time spent in wakefulness decreased by 6.09 min (*n* = 8, *p* <0.01) in DMH–vGAT–Casp3 mice. Compared with the control group, the amount of time spent in NREM sleep by DMH–vGAT–Casp3 mice during cold exposure decreased by 75.70%, and the time spent in wakefulness increased by 77.12% (Figure 6L).

The above results indicate that the selective apoptosis of DMH GABAergic neurons attenuates the wakefulness elicited by cold exposure and NREM sleep elicited by warm exposure.

### 2.4. Chemogenetic Activation of GABAergic Neurons in the DMH Increased Body Temperature during the Inactive Period but Did Not Influence Sleep–Wake Behaviors in vGAT–Cre Mice

To further study the roles of GABAergic neurons in the DMH in body temperature and sleep–wake regulation, we employed chemogenetics in combination with vGAT–Cre mice to specifically manipulate GABAergic neurons in the DMH (Figure 7A). We bilaterally microinjected the AAV–DIO–hM3Dq–mCherry in the DMH of vGAT–Cre mice (Figure 7B,C). Three weeks after AAV injection, we performed the locomotion activity, core body temperature, and EEG/EMG recordings of the DMH–vGAT–hM3Dq mice.

Immunohistochemistry showed that clozapine–N–oxide (CNO, 3 mg/kg), a specific hM3Dq agonist, but not saline, could drive c–Fos expression in hM3Dq–expressing neurons in the DMH (Appendix A). In addition, bath application of CNO (5 μM) depolarized the DMH hM3Dq-expressing neurons and significantly increased the firing of action potentials in hM3Dq/mCherry-positive neurons, as indicated by whole-cell current clamp recordings. Thus, the DREADD system used in this study stimulated the activity of DMH neurons both in vivo and in vitro (Figure 7D–H).

Compared with the vehicle groups, after the administration of CNO at 09:00, the core body temperature of DMH–vGAT–hM3Dq mice increased slightly for three hours (*n* = 6–7), and the average temperature increased by 0.16 °C (Figure 7I,J), but activation of the GABAergic neurons in the DMH did not influence the locomotor activity or the sleep–wake cycle (Figure 7K). Meanwhile, after the administration of CNO at 21:00, compared with the vehicle groups, the activation of the GABAergic neurons in the DMH did not influence the locomotion activity, the core body temperature, or the sleep–wake cycle (Appendix A).

### 2.5. Chemogenetic Inhibition of GABAergic Neurons in the DMH Decreased Body Temperature and Increased NREM Sleep in vGAT–Cre Mice during the Inactive Period

To investigate whether GABAergic neurons in the DMH play roles in temperature regulation and sleep–wake regulation, we also employed chemogenetics to inhibit the activity of these neurons (Figure 8A). We bilaterally microinjected AAV–DIO–hM4Di–mCherry into the DMH of vGAT–Cre mice (Figure 8B,C). Three weeks after AAV injection, we performed locomotion activity, core body temperature, and EEG/EMG recordings of DMH–vGAT–hM4Di mice. In addition, bath application of CNO (5 μM) reduced the spontaneous firing rate of GABAergic neurons expressing hM4Di receptors in the DMH as indicated by whole-cell current clamp recordings (Figure 8D–G). 

Compared with the vehicle groups, after the administration of CNO at 09:00 or 21:00, the core body temperature of DMH–vGAT–hM4Di mice was decreased for two hours (*n* = 7, *p* < 0.05) (Figure 8H,I).

Compared with the vehicle groups, after the administration of CNO at 09:00, REM sleep decreased by 68.02% (Figure 8M,P). Meanwhile, after the administration of CNO at 21:00, inactivation of the GABAergic neurons in the DMH did not influence the sleep–wake cycle (Appendix A).

## 3. Discussion

The DMH is involved in the regulation of a variety of body life activities, including feeding [17,27,28,29], food expectations [30], stress [19,31], metabolism [32,33], and locomotion activities [18]. In recent years, studies of the DMH in various functions have increased. The DMH has always been an important part of body temperature regulation, and body temperature will be affected by ambient temperatures. Shen et al. demonstrated that acute warm exposure does not affect the activity of DMD neurons, but acute cold exposure can activate GABAergic and glutamatergic neurons in the DMD. Furthermore, optogenetic activation of DMD neurons induces the increase of the core body temperature and brown fat production. On the contrary, optogenetic inactivation of the activity of these neurons can reduce heat production [24]. Similarly, Borbely et al. found that the elevation of ambient temperatures caused a significant rise in cortical temperature but not in hypothalamic temperature [34]. In our study, the apoptosis of DMH neurons or GABAergic neurons did not affect core body temperature regulation during acute cold exposure but increased core body temperature fluctuations—that is, the drop of the core body temperature after acute warm exposure. The inactivation of these neurons can decrease the physiological body temperature. These results are not entirely consistent with those mentioned above. The reason for these inconsistencies may be that the target area of our study does not completely coincide with that of others, and after the apoptosis of DMH neurons or GABAergic neurons, temperature regulation due to cold exposure may have a compensatory effect.

At the same time, ambient temperatures can induce changes in sleep–wake behavior [3,18,24]. Borbely’s team found that changes in ambient temperatures did not alter the sleep–wake cycle but caused a significant reduction of waking along with an increase in NREM sleep [21]. Meanwhile, Dan et al. found that galanin-expressing GABAergic neurons in the DMH comprise separate subpopulations with opposing effects on REM versus NREM sleep. Previous studies were conducted solely on the body temperature regulation or sleep–wake behavior that DMH participated in and did not clarify whether the process of body temperature regulation that DMH participated in would affect sleep–wake behavior.

In this study, neuronal apoptosis, chemogenetics, and in vitro electrophysiology were employed to explore the roles of DMH GABAergic neurons in sleep–wake behaviors elicited by ambient temperature. The results show that the apoptosis of DMH GABAergic neurons attenuated the change in sleep–wake behavior elicited by cold or warm exposure. The inactivation of these neurons can decrease REM sleep during the inactive period. These results suggest that GABA neurons in the DMH play a major role in the sleep–wake changes elicited by warm exposure, and that there may be interactions among the various neuron populations of the DMH in the sleep–wake changes elicited by cold exposure. 

In our study, the chemogenetic inactivation of DMH neurons caused a significant decrease in the core body temperature in the second and third hours after the administration of CNO, at which time the change in the core body temperature was transmitted to the center through the thermo-sensitive metabolic sensor in the body, and the thermoregulation was triggered by changes in metabolism. In general, the metabolic rate during REM sleep is relatively low, so the initiation of the thermoregulation process leads to a decrease in transient REM sleep. We speculate that this is a self-protection mechanism in stressful situations.

DMH is a heterogeneous nucleus containing orexin-expressing, glutamatergic, cholinergic [17], and GABAergic neurons, which can also be divided into many subtypes, such as galaninergic-expressing [23] and neuropeptide Y [35]. There are a large number of orexin neurons in the DMH [36], which also intensively project with the median preoptic nucleus and send projections to the ventrolateral preoptic nucleus, an important sleep regulation center [24], and the LH, the main arousal regulation center. Schmidt et al. found that the melanin-concentrating hormone system within the LH plays a critical role in REM sleep increases in warm ambient temperatures.

From an energetic viewpoint, one would think that the most appropriate temperature is at thermoneutrality, where the metabolic demands are minimal [37]. Mice are considered homeotherms, meaning that their body temperature is constant, or nearly so, even in front of important changes in ambient temperature [38]. Homoiothermy is strictly regulated by neural mechanisms. During cold exposure, the larger temperature gradient between the body core and shell causes peripheral vasoconstriction and thermolysis increases [39]. At this time, in order to maintain constant body temperature, thermogenesis increases, meaning the increase of the metabolism of the body. In fact, small changes in ambient temperature within the animal’s thermo-neutral zone do not cause significant changes in the metabolic rate. Cold or warm exposure outside the range of the thermo-neutral zone activates the thermoregulation mechanism of the body, which can lead to an increase in the metabolic rate.

In addition to the above roles of the DMH, it is also part of the important downstream of the biological rhythm center, the SCN [40]. The biological rhythm of the LC neurons disappears after the apoptosis of the DMH neurons [41]. In general, the core body temperature of the thermostatic animals will produce a certain oscillation curve according to the light/dark cycles [42]. The biological rhythm mainly controlled by the SCN is endogenous but it is also affected by the environmental cycles. There have been some studies that suggest that the circadian rhythm of locomotor activity in mice and dromedary camels is affected by the ambient temperature cycle [43,44]. Although, the effect of the ambient temperature cycle is weaker than the light/dark cycle [44], which are affected by the range, amplitude, and time of the ambient temperature cycle. The GABA_A_ receptor and glutamate decarboxylase mRNA in the DMH change along with biological rhythm [45], which also suggests that GABAergic neurons are involved in the regulation of biological rhythms. Early studies showed that damage to the DMH with ibotenic acid (IBO) disturbs the biological rhythm of rats’ wakefulness, feeding, locomotion activity, serum corticosterone secretion, and brain temperature [41]. However, in our research, the destruction of neurons in the DMH did not cause similar changes in mice. There may be two possible reasons for these seemingly contradictory results. First, the previous method of IBO damage was not specific; second, IBO is chemotoxic, and there may be functional compensation during the period. By contrast, the effect of chemogenetic inhibition on neurons has higher time precision. It is interesting that our study found that, after the activation or inhibition of the GABAergic neurons in the inactive period of mice, the increase or decrease in core body temperature of the mice was significantly higher than the changes in the active period, which is also auxiliary evidence that the DMH neurons participate in the regulation of biological rhythms.

In this study, we performed cold exposure during the light period when mice are mostly sleepy to better observe the wake-promoting effect, while, in the dark period when the mice were more active, warm exposure was implemented to more clearly observe the sleep-promoting effect. We hypothesize that if the warm exposure occurs during the light period (inactive phase) of mice, which caused the core body temperature to become lower than the acral temperature, this may be more conducive to be sleep promotion. However, in the dark period (active phase) of mice, cold exposure will make the acral temperature lower than the core body temperature, which may induce heat redistribution and induce an arousal state. This reverse study design may not change the sleep or wakefulness state of cold/warm exposure during dark/light periods.

Our research can provide a direction for using ambient temperatures to improve sleep quality, and external ambient temperatures can be extended to the body’s local/micro ambient temperature environment [22,46]. Therefore, DMH is a potential target for sleep regulation with temperature and biological rhythm regulation.

## 4. Materials and Methods

### 4.1. Animals

Male, specific pathogen free (SPF), inbred C57BL/6J mice (10–14 weeks old weighing 20–25 g) were obtained from Shanghai Xipuer–Bikai Laboratory Animal Co. Ltd (Shanghai, China). Adult male (8–16 weeks old, 22–30g) vGAT (Slc32al or Viaat)–Cre mice with a C57BL/6J background in which Cre–recombinase was expressed under the promoter of the vesicular GABA transporter (vGAT) gene were purchased from Jackson Laboratory (stock number: 017535; Bar Harbor, ME, USA). The mice were housed at a constant temperature (22 ± 0.5 °C) and humidity (60 ± 2%) under an automatically controlled 12/12 h light/dark cycle (lights on at 7:00 a.m., illumination intensity ≈100 lux). Food and water were available ad libitum. Every effort was made to minimize animal suffering, and the minimum number of animals required to generate reliable scientific data was used. 

### 4.2. AAV Vectors

AAV–hSyn–DIO–hM3Dq–mCherry and AAV–hSyn–DIO–hM4Di–mCherry were generated by tripartite transfection (AAV2/10 expression plasmid, adenovirus helper plasmid, and pAAV plasmid) into 293A cells. The final viral concentrations of the transgenes were 1–2 × 10^12^ genome copies/ mL (Wuhan BrainVTA Co. Ltd., Wuhan, China). An AAV vector carrying the hSyn–cre, EF1a–DIO–taCasp3–TEVp–WPRE–PA, EF1a–DIO–eGFP–WPRE–pA construct was packaged into an AAV2/9 serotype with titers 1–2 × 10^12^ genome copies/ mL (Shanghai Taiting Biological Co., Ltd. Shanghai, China). Aliquots of viral vectors were stored at −80 °C before stereotaxic injection. 

### 4.3. Stereotaxic Surgery

All mice used were anesthetized with pentobarbital sodium (50 mg/kg, i.p.) for surgical procedures and placed into a stereotactic frame (RWD Life Science, Shenzhen, China). A burr hole was made, and a fine glass pipette (15–20 μm tip) containing recombinant AAV–hM3Dq/ hM4Di and AAV–Caspase3 virus was inserted bilaterally into the DMH (anteroposterior (AP): −0.9 mm, mediolateral (ML): ± 0.3 mm, dorsoventral (DV): −5.0 mm). A total of 30–40 nL of the virus was delivered to each site over a 5 min period via nitrogen gas pulses of 20–40 psi using an air compression system (Picospritzer III, Parker Hannifin Corp., Cleveland, OH, USA), and the needle was left in place for at least 10 min to permit diffusion as previously described. Mice that received bilateral injections were used for all experiments and received additional surgical implants after viral injection as described below. Only data from mice in which the infection area was confirmed were accepted.

At 14 days after AAV injection, mice were implanted with EEG and EMG electrodes for polysomnographic recordings. EEG and EMG signals were recorded from stainless steel screws inserted in the skull and 2 flexible silver wires inserted in the neck muscle, respectively. 

Then, 7 days later, brains were processed for fluorescence imaging or prepared for in vitro electrophysiological studies. 

### 4.4. The Intraperitoneal Implantation of DSI Telemetry Implant in Mice and the Control of the Constant Temperature Hot–Plate 

Before surgery, the DSI telemetry implants (DSI, St. Paul, MN, USA) were immersed in 75% alcohol to disinfect. Mice were anesthetized with pentobarbital sodium (50 mg/kg, i.p.), and the hair in the abdominal surgery area was shaved. The skin was disinfected with 75% alcohol. The skin and abdominal wall muscles were exposed layer by layer. The telemetry implant was placed in the abdominal cavity of the mouse, and the small ring implanted in the outer layer was sutured on the abdominal wall muscle of the mouse using a 5.0 suture to fix the mouse. The wound was sutured layer by layer, and penicillin (80,000 units/kg) was injected into the cavity to prevent infection. After 7–10 days of recovery, the core temperature/locomotion activity recording experiment was performed.

We used a customized U-shaped hollow heat-conducting aluminum alloy plate and a DC0506 low-temperature thermostatic bath with an external circulation and compressor as the thermostatic control system. The two were connected by a silicone hose. The two low-temperature thermostatic baths were set at two different experimental temperatures according to the experimental requirements. By switching the source of the water flowing through the thermally conductive aluminum plate between the two thermostatic baths, temperature conversion can be achieved, and the accuracy reached the minute level [47]. 

### 4.5. Polygraphic Recordings and Analysis

After 2–3 weeks for postoperative recovery and transgene expression, the animals were housed individually in recording chambers and connected to the EEG/EMG head stages. The recording cable was attached to a slip-ring unit so that the movement of mice would not be restricted. Mice were habituated to recording cables for 3–4 days before starting recording.

The EEG/EMG signal was recorded under baseline (free-moving, 28 ± 0.5 °C) and different treatment conditions over several days.

For DREADD experiments, all mice received vehicle or CNO (3 mg/kg, C2041, LKT Laboratories, St. Paul, MN, USA) treatment on 2 consecutive days at 09:00 (inactive period) or 21:00 (active period), separated by a 3 day wash-out period. After recording, mice were killed 2 h after vehicle or CNO treatment and then used for immunohistochemical staining. The EEG/EMG signal was amplified, bandpass filtered (EEG, 5–30 Hz; EMG, 40–200 Hz), digitized at 128 Hz, and recorded with Vital Recorder software (Kissei Comtec Co., LTD, Tokyo, Japan). The sleep state was scored using sleep analysis software (SleepSign, Kissei Comtec). All scoring was automatic based on EEG and EMG waveforms in 4 s epochs for chemogenetics. We defined wakefulness as desynchronized EEG and high levels of EMG activity, NREM sleep as synchronized, high-amplitude, low-frequency (0.5–4 Hz) EEG signals in the absence of motor activity, and REM sleep as having pronounced theta like (4–9 Hz) EEG activity and muscle atonia. Vigilance states assigned by SleepSign (Kissei Comtec Co., LTD, Tokyo, Japan) were examined visually and corrected manually if necessary [48,49]. 

### 4.6. Locomotor Activity and Core Temperature Recordings

In this study, we used the implantable physiological signal wireless telemetry system produced by the American DSI Company to record the long-term core body temperature and locomotor activity of free-moving mice, which can better reflect the physiological conditions of mice. The system consists of an implant, a receiver, a data converter (DEM), and a data analysis computer (Dataquest ART). The 1 cm^2^ × 0.5 mm implant integrated a sensor, amplifier, and wireless signal transmitter. The TA–F10 implant was embedded in the abdominal cavity of a mouse. The collected temperature signal was converted into a radio signal and received by a receiver placed under the recording cage. The signal of locomotor activity was directly transmitted by the cross in the receiver with an antenna for monitoring. The signal was converted by a data-converter into a computer for processing [47]. 

### 4.7. In Vitro Electrophysiology

At 3–4 weeks after AAV–hM3Dq/ hM4Di injections, vGAT–Cre mice were anesthetized and perfused transcardially with ice-cold modified aCSF saturated with 95% O_2_ and 5% CO_2_ and containing (in mM): 215 sucrose, 26 NaHCO_3_, 10 glucose, 3 MgSO_4_, 2.5 KCl, 1.25 NaH_2_PO_4_, 0.6 mM Na pyruvate, 0.4 ascorbic acid, and 0.1 CaCl_2_. Brains were then rapidly removed, and acute coronal slices (300 μm) containing the DMH were cut on a vibratome (VT1200, Leica, Weztlar, Germany) in ice-cold modified aCSF. Next, slices were transferred to a holding chamber containing normal recording aCSF (in mM): 125 NaCl, 26 NaHCO_3_, 25 glucose, 2.5 KCl, 2 CaCl_2_, 1.25 NaH_2_PO_4_ and 1.0 MgSO_4_, and allowed to recover for 30 min at 32 °C. Then, slices were maintained at room temperature (RT) for 30 min before recording. During recording, slices were submerged in a recording chamber superfused with aCSF (2 mL/min) at 30–32 °C. Slices were visualized using a fixed-stage upright microscope (BX51W1, Olympus Corporation, Tokyo, Japan) equipped with a 40× water immersion objective and an infrared-sensitive CCD camera. Expression of hM3Dq/ hM4Di was confirmed by visualization of mCherry fluorescence in GABAergic neurons. Patch pipettes were fabricated from thick-walled borosilicate glass capillaries (1.5 mm outer diameter, 0.86 mm internal diameter, Vital Sense; Scientific Instruments Co., Ltd., Bangkok, Thailand) using a 2-step vertical puller (PC–10, Narishige, Tokyo, Japan) and had resistances between 4 and 6 MΩ. Recording pipettes were filled with an internal solution containing (in mM): 105 potassium gluconate, 30 KCl, 10 phosphocreatine, 4 ATP–Mg, 0.3 EGTA, 0.3 GTP–Na, and 10 HEPES (pH 7.3, 285–300 mosm). Recordings were conducted in the whole-cell or cell-attached configuration using a Multiclamp 700B amplifier (Axon Instruments, Union City, CA, USA). Signals were filtered at 4 kHz and digitized at 10 kHz with a DigiData 1440A (Axon Instruments, Union City, CA, USA). Data were acquired and analyzed with pClamp10.3 software (Axon Instruments, Union City, CA, USA) [50]. 

### 4.8. Immunohistochemistry

After whole-cell recording, slices containing biocytin-loaded cells were fixed in 4% paraformaldehyde (PFA), and nonspecific binding was blocked with 5% donkey serum in phosphate-buffered saline (PBS). Then, slices were incubated overnight at 4 °C in PBS containing 0.3% Triton–X (PBST). For double immunostaining of c–Fos and mCherry, after vehicle or CNO administration, mice were deeply anesthetized by pentobarbital sodium (50 mg/kg, i.p.) and then perfused intracardially with 30 mL PBS followed by 30 mL 4% PFA. Brains were removed, postfixed for 6 h in 4% PFA, and then incubated in 30% sucrose phosphate buffer (PB) at 4 °C until they sank. Coronal sections (30 μm) were cut on a freezing microtome (CM1950, Leica, Germany) in 4 series. The floating sections were washed in PBS and incubated with a rabbit polyclonal antibody against c–Fos (1:10,000, pc–38, Calbiochem) in PBST for 48 h at 4 °C on an agitator. After washing, sections were incubated with a biotinylated goat anti-rabbit IgG antibody (1:1000, BA–1000, Vector Laboratories, Burlingame, CA, USA) followed by incubation in an avidin–biotin peroxidase complex (ABC) solution (1:1000, PK–6100, Vector Laboratories) for 1 h. After rinsing, the sections were immersed in a 3,3–diaminobenzidine–4 HCl (DAB) and nickel solution (SK–4100, Vector Laboratories) for 5–10 min at RT, in which cFos-immunoreactive neurons were identified by the presence of black reaction products. The following day, the c–Fos immunostained sections were incubated in a rabbit polyclonal antibody against DsRed (mCherry tag, 1:5000, 632496, Clontech, Mountain View, CA, USA) overnight at 4 °C. Amplification steps were similar to those described above, except that the last step was performed in a DAB solution without nickel ammonium sulfate. Finally, the sections were mounted on glass slides, dried, dehydrated, and cover-slipped. 

### 4.9. Statistical Analysis

Data are presented as mean ± S.E.M. Sample sizes were chosen based on previous studies. Repeated ANOVA was used to perform group comparisons with multiple measurements. Two-tailed paired Student’s *t*-test was used for single value comparisons. One-way ANOVA was used to compare more than two groups. Two-way ANOVA followed by the probable least-squares difference test was used for group comparisons with two factors. *p* values less than 0.05 were considered statistically significant. Statistical analysis was performed using SPSS 23.0 software. 

## 5. Conclusions

Cold exposure increased wakefulness while warm exposure increased NREM sleep in WT mice. Apoptosis of the neurons, including GABAergic neurons, in the DMH attenuated sleep and wake changes was elicited by cold or warm exposure. Therefore, the GABAergic neurons in the DMH are involved in sleep–wake regulation elicited by ambient temperatures.

## Figures and Tables

**Figure 1 ijms-23-01270-f001:**
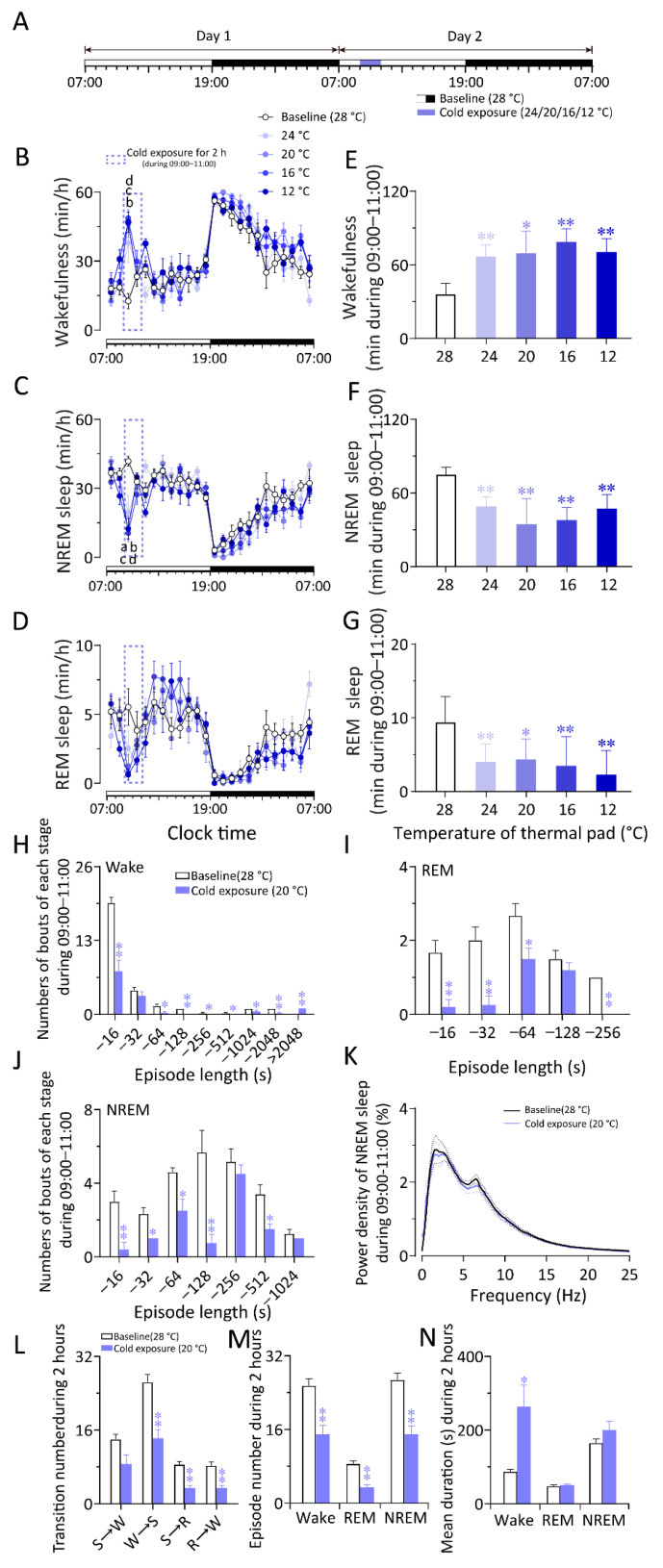
Cold exposure increased wakefulness and decreased NREM and REM sleep in WT mice. (**A**) Schematic diagram of sleep recording and cold exposure. (**B**–**D**) 24 h time–course changes of wakefulness (**B**), NREM (**C**)and REM sleep (**D**) in WT mice exposed to a 2 h cold ambient temperature (24 °C, 20 °C, 16 °C, 12 °C, respectively) or constant 28 °C at 09:00 (light phase). (**E**–**G**) Total time spent during 09:00–11:00 of wakefulness (**E**), NREM (**F**), and REM sleep (**G**) in WT mice exposed to a 2 h cold ambient temperature (24 °C, 20 °C, 16 °C, 12 °C, respectively) or constant 28 °C at 09:00 (light phase). (**H**–**J**) Numbers of bouts of wakefulness (**H**), REM sleep (**I**), and NREM sleep (**J**) during cold exposure (09:00–11:00). (**K**) Power density of NREM sleep during cold exposure (09:00–11:00). The power of each 0.25 Hz bin was averaged and normalized by calculating the percentage of each bin from the total power (0–25 Hz). (**L**) Transition number of each stage during cold exposure (09:00–11:00). S: NREM sleep; W: Wake; R: REM sleep. (**M**) Episode number of each stage during cold exposure (09:00–11:00). (**N**) Mean duration of each stage during cold exposure (09:00–11:00). Values are means ± SEM (*n* = 6–7). ^a^
*p* < 0.01 indicated significant differences between the 24 °C group from the baseline (28 °C) group, ^b^
*p* < 0.01 indicated significant differences between the 20 °C group from the baseline (28 °C) group, ^c^
*p* < 0.01 indicated significant differences between the 16 °C group from the baseline (28 °C) group, ^d^
*p* < 0.01 indicated significant differences between the 12 °C group from the baseline (28 °C) group as assessed by repeated ANOVA (B and C); * *p* < 0.05 and ** *p* < 0.01 indicated significant differences from the baseline (28 °C) group as assessed by one-way ANOVA (**E**–**G**) and two-tailed paired Student’s *t*-test (**H**–**N**).

**Figure 2 ijms-23-01270-f002:**
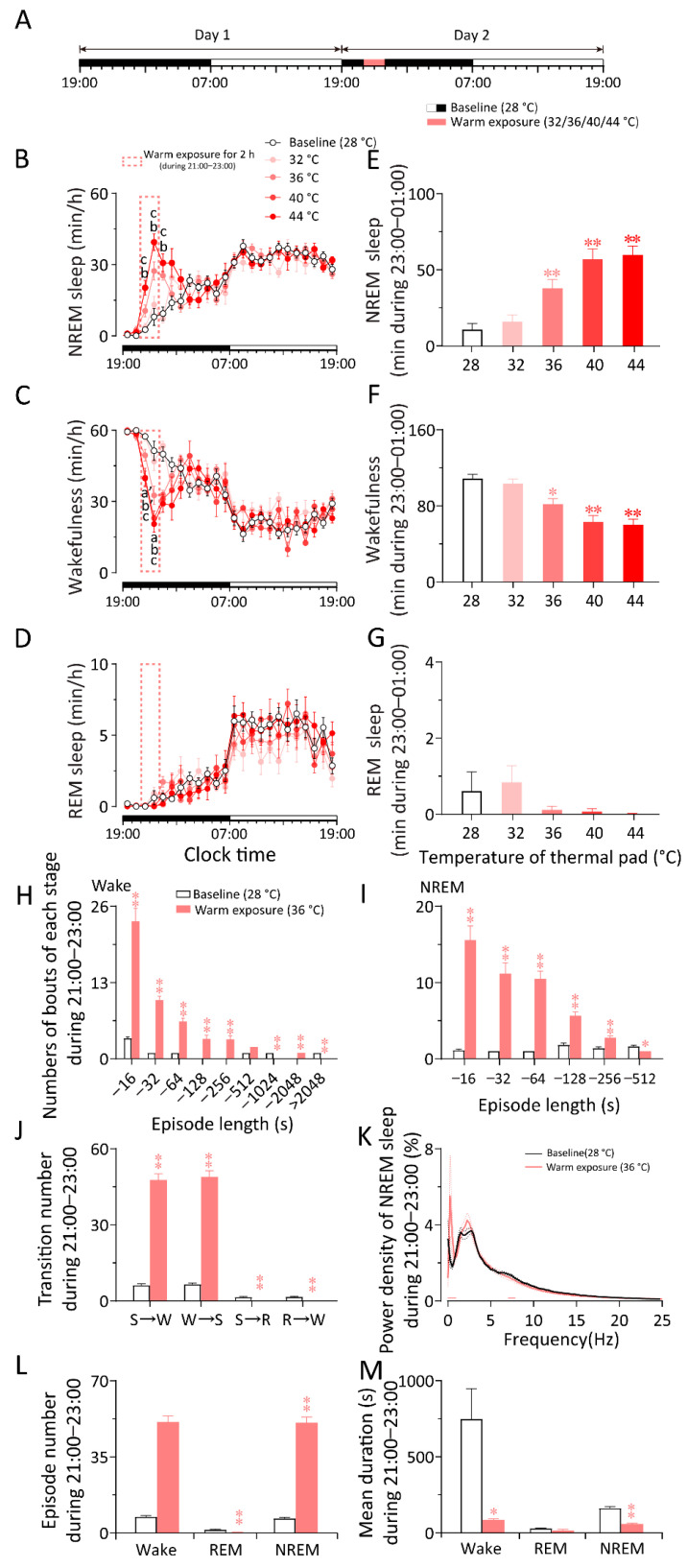
Warm exposure increased NREM sleep and decreased wakefulness in WT mice. (**A**) Schematic diagram of sleep recording and warm exposure. (**B**–**D**) 24 h time–course changes of NREM sleep (**B**), wakefulness (**C**), and REM sleep (**D**) in WT mice exposed to a 2 h warm ambient temperature (32 °C, 36 °C, 40 °C, 44 °C, respectively) or constant 28 °C at 21:00 (dark phase). (**E**–**G**) Total time spent during 21:00–23:00 of NREM sleep (**E**), wakefulness (**F**), and REM sleep (**G**) in WT mice exposed to a 2 h warm ambient temperature (32 °C, 36 °C, 40 °C, 44 °C, respectively) or constant 28 °C at 21:00 (dark phase). (**H**) Numbers of bouts of wakefulness during warm exposure (21:00–23:00). (**I**) Numbers of bouts of NREM sleep during warm exposure (21:00–23:00). (**J**) Transition number of each stage during warm exposure (21:00–23:00). S: NREM sleep; W: Wake; R: REM sleep. (**K**) Power density of NREM sleep during warm exposure (21:00–23:00). The power of each 0.25 Hz bin was averaged and normalized by calculating the percentage of each bin from the total power (0–25 Hz). The horizontal bars indicated statistical differences (*p* < 0.05). (**L**) Episode number of each stage during warm exposure (21:00–23:00). (**M**) Mean duration of each stage during warm exposure (21:00–23:00). Values are means ± SEM (*n* = 6–7). ^a^
*p* < 0.01 indicated significant differences between the 36 °C group from the baseline (28 °C) group, ^b^
*p* < 0.01 indicated significant differences between the 40 °C group from the baseline (28 °C) group, ^c^
*p* < 0.01 indicated significant differences between the 44 °C group from the baseline (28 °C) group, ^a’^*p* < 0.05 indicated significant differences between the 36 °C group from the baseline (28 °C) group, ^b’^
*p* < 0.05 indicated significant differences between the 40 °C group from the baseline (28 °C) group as assessed by repeated ANOVA (B and C); * *p* < 0.05 and ** *p* < 0.01 indicated significant differences from the baseline (28 °C) group as assessed by one-way ANOVA (**E**–**G**) and two-tailed paired Student’s *t*-test (**H**–**M**).

**Figure 3 ijms-23-01270-f003:**
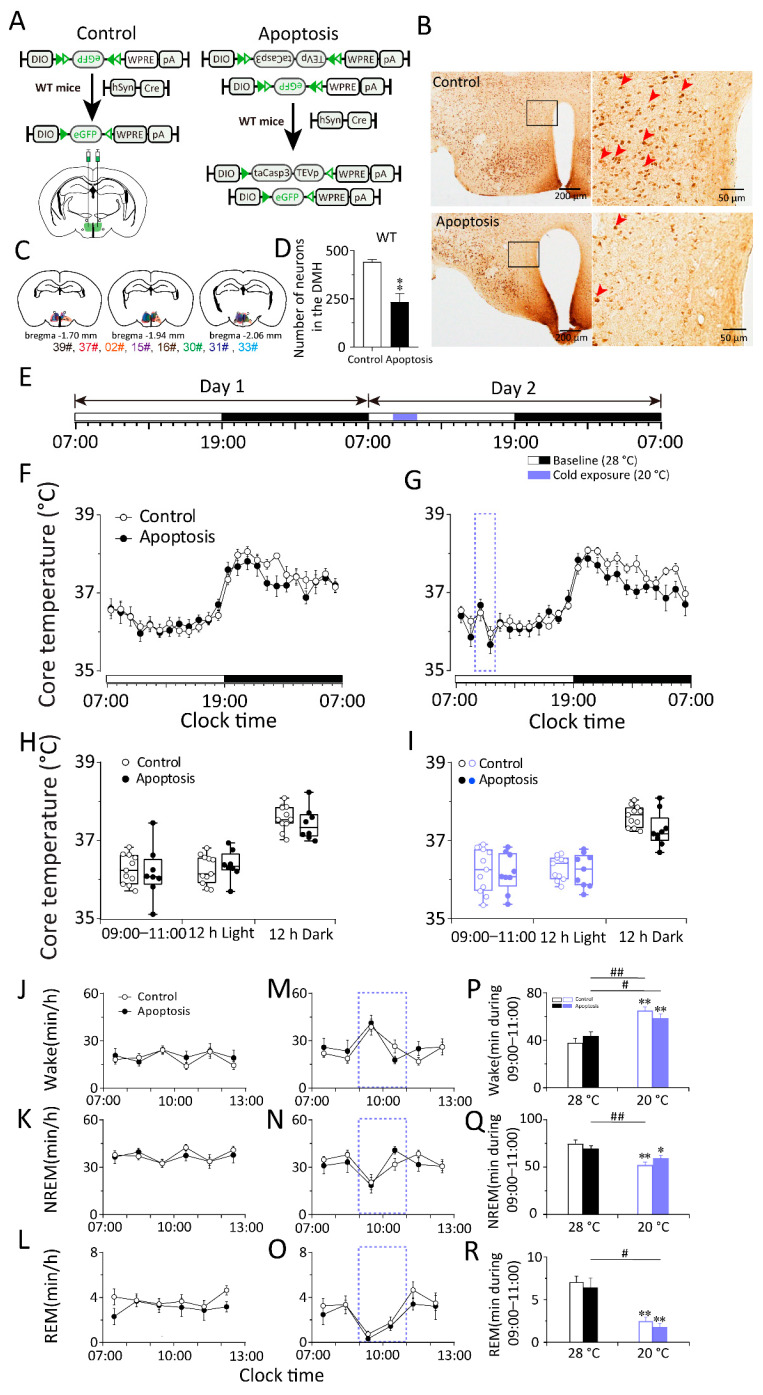
Cold exposure did not influence the core body temperature and sleep–wake cycle of co-trol or DMH apoptosis group in WT mice. (**A**) The schematic representation of nonspecific apoptosis of DMH neurons in mice. AAV–DIO–taCasp3–TEVp, AAV–DIO–eGFP, and AAV–hSyn–Cre were microinjected into the DMH of apoptosis mice. AAV–DIO–eGFP and AAV–hSyn–Cre were microinjected into the DMH of control mice. AAV: Adeno-associated virus; DIO: Double-floxed inverse orientation; hSyn: Human synapsin; eGFP: Enhanced green fluorescent protein; TEVp: Tobacco etch virus protease; WPRE: Woodchuck hepatitis virus posttranscriptional regulatory element; pA: poly(A) tail. (**B**) Immunohistochemical staining of NeuN in typical brain sections from control and apoptosis mice. Left panels were immunohistochemical diagram of low magnification views including DMH. Right panels indicated high magnification views of rectangular areas within the DMH marked in left panels. The red arrows indicated typical positive signal of NeuN. The positive NeuN expression in the DMH and intact ventriculus tertius remained in control mice. There was little NeuN expression in the DMH of apoptosis mice. The upper part of the ventriculus tertius was expanded. Scale bars in the left panel: 200 μm; in the right panel: 50 μm. NeuN: Neuronal Nuclei. (**C**) Drawings of superimposed AAV microinjection sites in the DMH of apoptosis mice (*n* = 8, indicated with different colors). (**D**) Numbers of neurons in the DMH in control and apoptosis mice. (**E**) The schematic representation of cold exposure. The 12 h light period (indicated as open bars) was followed by the 12h dark period (indicated as filled bars) in two consecutive days with the baseline temperature of 28 °C. The blue band represented a 2 h 20 °C cold exposure period which started at 9:00. (**F**,**G**) 24 h time–course of the core body temperature of baseline day (**F**) and exposure day (**H**) between control and apoptosis group. (**H**,**I**) The average value of the core body temperature during 9:00–11:00, 12 h Light and 12 h Dark in baseline day (**G**) and exposure day (**I**) between control and apoptosis group. (**J**–**R**) The left and middle column: 6 h time–courses of the wakefulness (**J**,**M**), NREM sleep (**K**,**N**), and REM sleep (**L**,**O**) during 7:00–13:00 in baseline day and cold exposure day between control and apoptosis group. The right column (**P**,**Q**,**R**): average values of each stage during 9:00–11:00 in baseline day and cold exposure day between control and apoptosis group. Values were presented as means ± S.E.M (*n* = 8–11). ** *p* < 0.01 indicated significant differences from the control mice as assessed by one-way ANOVA (**D**); * *p* < 0.05 and ** *p* < 0.01 indicated significant differences from the baseline (28 °C)-control mice; # *p* < 0.05 and ## *p* < 0.01 indicated significant differences from the baseline (28 °C)-apoptosis mice as assessed by two-way ANOVA followed by the probable least-squares difference test (**P**–**R**).

**Figure 4 ijms-23-01270-f004:**
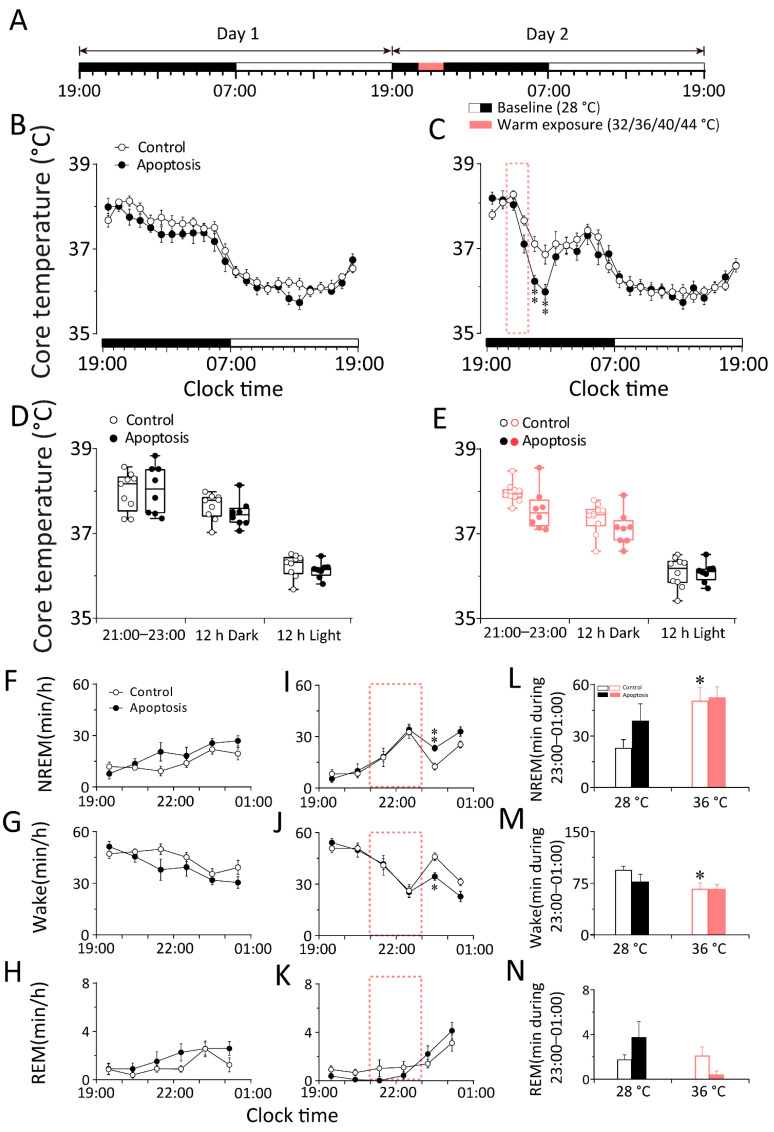
The apoptosis of DMH neurons increased the core temperature fluctuation elicited by warm exposure and alleviated the NREM sleep elicited by warm exposure in WT mice. (**A**) The schematic representation of warm exposure. The 12 h light period (indicated as open bars) was followed by the 12 h dark period (indicated as filled bars) in two consecutive days with the baseline temperature of 28 °C. The red band represented the 2 h 36 °C warm exposure period which started at 21:00. (**B**–**E**) 24 h time–courses of the core body temperature of in baseline day (**B**) and exposure day (**C**) between control and apoptosis group. The average value of the core body temperature during 21:00–23:00, 12 h dark and 12 h light in baseline day (**D**) and exposure day (**E**) between control and apoptosis group. (**F**–**N**) The left and middle column: 6 h time–courses of the NREM sleep (**F**,**I**), wakefulness (**G**,**J**), and REM sleep (**H**,**K**) during 19:00–01:00 in baseline day and warm exposure day between control and apoptosis group. The right column (**L**–**N**): average values of each stage during 21:00–23:00 in baseline day and cold exposure day between control and apoptosis group. Values were presented as means ± S.E.M (*n* = 8–11). ** *p* < 0.01, * *p* < 0.05 indicated significant differences from the control group as assessed by repeated ANOVA measurement (**C**,**I**,**J**); * *p* < 0.05 indicated significant differences from the baseline (28 °C)-control mice as assessed by two-way ANOVA followed by the probable least-squares difference test (**L**,**M**).

**Figure 5 ijms-23-01270-f005:**
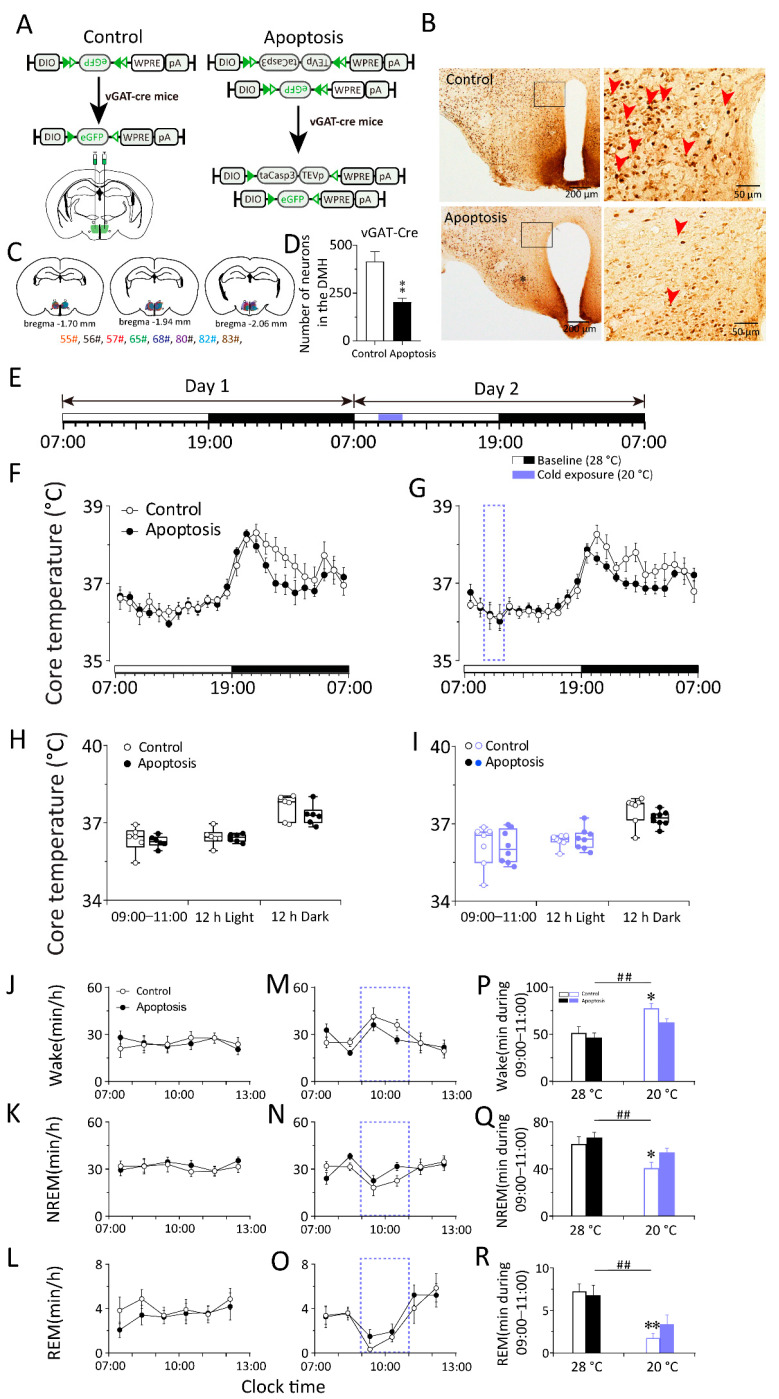
Cold exposure did not influence the core body temperature of the control or DMH GABAergic apoptosis mice, whereas it alleviated the wakefulness elicited by cold exposure in vGAT-Cre mice. (**A**) The schematic representation of nonspecific apoptosis of DMH GABAergic neurons in mice. AAV–DIO–taCasp3–TEVp and AAV–DIO–eGFP were microinjected into the DMH of apoptosis mice. AAV–DIO–eGFP were microinjected into the DMH of control mice. (**B**) Immunohistochemical staining of NeuN in typical brain sections from control and apoptosis mice. Left panels were immunohistochemical diagram of low magnification views including DMH. Right panels indicated high magnification views of rectangular areas within the DMH marked in left panels. The red arrows indicated typical positive signal of NeuN. The positive NeuN expression in the DMH and intact ventriculus tertius remained in control mice. There was little NeuN expression in the DMH of apoptosis mice. The upper part of the ventriculus tertius was expanded. Scale bars in the left panel: 200 μm; in the right panel: 50 μm. (**C**) Drawings of superimposed AAV microinjection sites in the DMH of apoptosis mice (*n* = 8, indicated with different colors). (**D**) Numbers of neurons in the DMH in vGAT–Cre mice. (**E**) The schematic representation of cold exposure. The 12 h light period (indicated as open bars) was followed by the 12 h dark period (indicated as filled bars) in two consecutive days with the baseline temperature of 28 °C. The blue band represented a 2 h 20 °C cold exposure period which started at 9:00. (**F**–**I**) 24 h time–course of the core body temperature of baseline day (**F**) and exposure day (**G**) between control and apoptosis group. The average value of the core body temperature during 9:00–11:00, 12 h Light and 12 h Dark in baseline day (**H**) and exposure day (**I**) between control and apoptosis group. (**J**–**R**) The left and middle column: 6 h time–courses of the wakefulness (**J**,**M**), NREM sleep (**K**,**N**), and REM sleep (**L**,**O**) during 7:00–13:00 in baseline day and cold exposure day between control and apoptosis group. The right column (**P**,**Q**,**R**): average values of each stage during 9:00–11:00 in baseline day and cold exposure day between control and apoptosis group. Values were presented as means ± S.E.M (*n* = 8–11). ** *p* < 0.01 indicated significant differences from the control mice as assessed by one-way ANOVA (D); * *p* < 0.05 and ** *p* < 0.01 indicated significant differences from the baseline (28 °C)-control mice; ## *p* < 0.01 indicated significant differences from the baseline (28 °C)-apoptosis mice as assessed by two-way ANOVA followed by the probable least-squares difference test (**P**–**R**).

**Figure 6 ijms-23-01270-f006:**
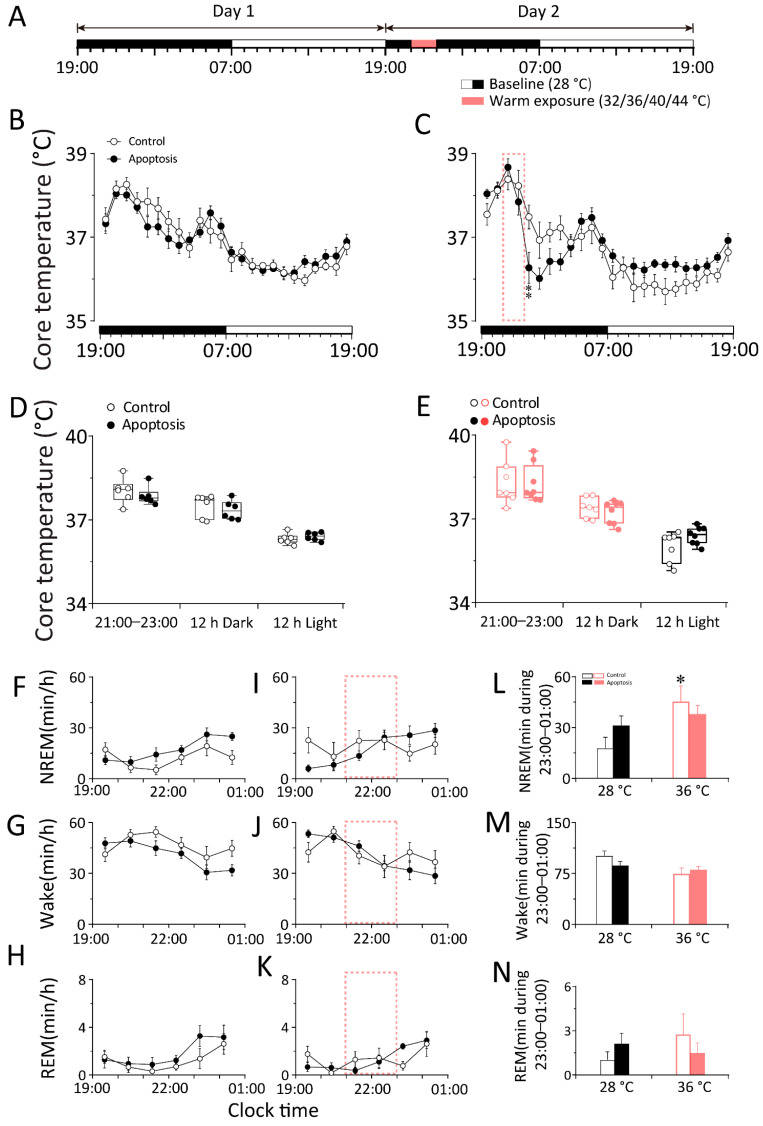
The apoptosis of DMH GABAergic neurons increased the core temperature fluctuation elicited by the warm exposure and alleviated the NREM sleep elicited by the warm exposure in vGAT–Cre mice. (**A**) The schematic representation of warm exposure. The 12 h light period (indicated as open bars) was followed by the 12 h dark period (indicated as filled bars) in two consecutive days with the baseline temperature of 28 °C. The blue band represented the 2 h 36 °C cold exposure period which started at 21:00. (**B**–**E**) 24 h time-courses of the core body temperature of in baseline day (**B**) and exposure day (**C**) between the control and apoptosis groups. The average value of the core body temperature during 21:00–23:00, 12 h dark and 12 h light in baseline day (**D**) and exposure day (**E**) between the control and apoptosis groups. (**F**–**N**) The left and middle column: 6 h time–courses of the NREM sleep (**F**,**I**), wakefulness (**G**,**J**), and REM sleep (**H**,**K**) during 19:00–01:00 in baseline day and warm exposure day between the control and apoptosis groups. The right column (**L**–**N**): average values of each stage during 21:00–23:00 in baseline day and warm exposure day between control and apoptosis group. Values were presented as means ± S.E.M (*n* = 8–11). ** *p* < 0.01 indicated significant differences from the control group as assessed by repeated ANOVA measurement (**C**); * *p* < 0.05 indicated significant differences from the baseline (28 °C)-control mice as assessed by two-way ANOVA followed by the probable least-squares difference test.

**Figure 7 ijms-23-01270-f007:**
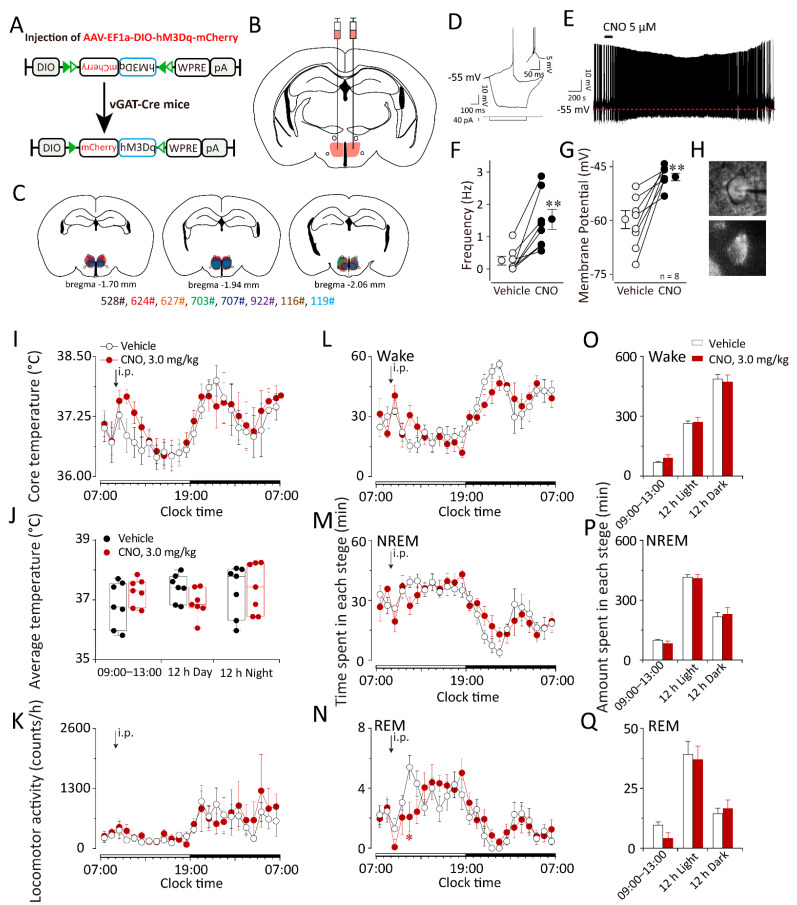
Chemogenetic activation of DMH GABAergic neurons did not affect the core temperature or sleep–wake cycle in vGAT–Cre mice. (**A**) The schematic representation of the microinjection of AAV–DIO–hM3Dq–mCherry into the DMH of vGAT–Cre mice. EF1α, elongation factor 1–alpha; DIO, double–floxed inverse orientation; hM3Dq, selective human muscarinic acetylcholine M3 receptor; WPRE, woodchuck hepatitis virus post-transcriptional regulatory element; pA, poly(A) tail. (**B**,**C**) Drawings of superimposed AAV microinjection sites in the DMH of vGAT–Cre mice (*n* = 8, indicated with different colors). (**D**) The typical electrophysiological trace showing the whole-cell current clamp recording of a GABAergic neuron expressing hM3Dq–mCherry fusion protein. (**E**) The administration of CNO (5 μM) elicited the rapid depolarization of the membrane potential and concomitantly increased the frequency of action potential firings. (**F**) The frequency of action potential firing before and after the administration of CNO (5 μM). (**G**) The membrane potential before and after the administration of CNO (5 μM). (**H**) The recorded neuron in phase contrast (the upper panel) and fluorescent (the lower panel) microscopes in one coronal brain section of a vGAT–Cre mouse expressing hM3Dq in the DMH. (**I**) Time–courses of the core body temperature of vGAT–Cre mice expressing hM3Dq in the DMH after the administration of vehicle or CNO (3 mg/kg) at 9:00. (**J**) Average values of the core body temperature during 4 h after the administration of vehicle or CNO (3 mg/kg) at 9:00. (**K**) Time–course of the locomotion after the administration of vehicle or CNO (3 mg/kg) at 9:00. (**L**–**N**) Time–courses of each stage of vGAT–Cre mice expressing hM3Dq in the DMH after the administration of vehicle or CNO (3 mg/kg) at 9:00. (**O**–**Q**) Amounts of each stage during 3 h, 12 h light or 12 h dark after the administration of vehicle or CNO (3 mg/kg). Values were presented as means ± S.E.M (*n* = 7–8). * *p* < 0.05 and ** *p* < 0.01 indicated significant differences from the vehicle group as assessed by one-way ANOVA (**F**,**G**) and repeated ANOVA (**N**).

**Figure 8 ijms-23-01270-f008:**
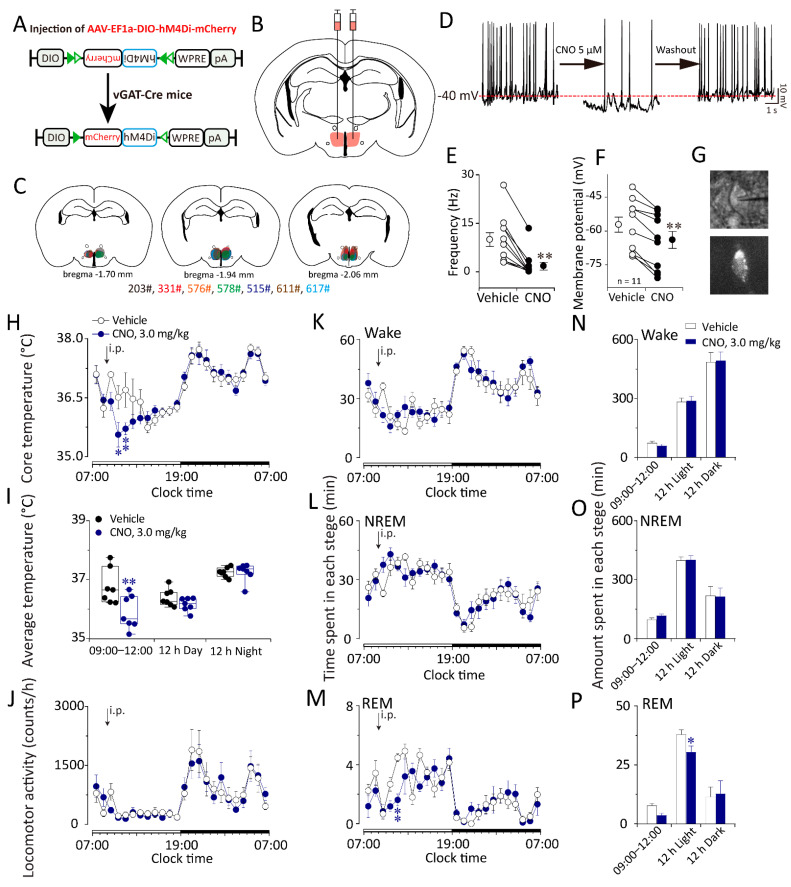
Chemogenetic inhibition of DMH GABAergic neurons lower the core temperature but did not affect sleep–wake cycle in vGAT–Cre mice. (**A**) The schematic representation of the microinjection of AAV–DIO–hM4Di–mCherry into the DMH of vGAT–Cre mice. hM4Di: selective human muscarinic acetylcholine M4 receptor. (**B**,**C**) Drawings of superimposed AAV microinjection sites in the DMH of vGAT–Cre mice (*n* = 8, indicated with different colors). (**D**) Bath application of CNO (5 μM) reduced the spontaneous firing rate of GABAergic neurons expressing hM4Di receptors in the DMH. (**E**) The frequency of action potential firing before and after the administration of CNO (5 μM). (**F**) The membrane potential before and after the administration of CNO (5 μM). (**G**) The recorded neuron in phase contrast (the upper panel) and fluorescent (the lower panel) microscopes in one coronal brain section of a vGAT–Cre mouse. (**H**) Time–course of the core body temperature of vGAT–Cre mice expressing hM4Di in the DMH after the administration of vehicle or CNO (3 mg/kg) at 9:00. (**I**) Average values of the core body temperature during 4 h after the administration of vehicle or CNO (3 mg/kg) at 9:00. (**J**) Time–course of the locomotion after the administration of vehicle of CNO (3 mg/kg) at 9:00. (**K**–**M**) Time–courses of each stage of vGAT–Cre mice expressing hM4Di in the DMH after the administration of vehicle or CNO (3 mg/kg) at 9:00. (**N**–**P**) Amounts of each stage during 3 h, 12 h light or 12 h dark after the administration of vehicle or CNO (3 mg/kg). Values were presented as means ± S.E.M (*n* = 7–8). * *p* < 0.05 and ** *p* < 0.01 indicated significant differences from the vehicle group as assessed by one-way ANOVA (**E**,**F**), repeated ANOVA (H and M) and two-tailed paired Student’s *t*-test (**I**,**P**).

## Data Availability

The data that support the findings of this study are available from the corresponding authors, Yi-Qun Wang and Zhi-Li Huang, upon reasonable request.

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
