# Peer review of "Role of Dorsomedial Hypothalamus GABAergic Neurons in Sleep–Wake States in Response to Changes in Ambient Temperature in Mice"

_ijms, 2022, doi:10.3390/ijms23031270_

Round 1

Reviewer 1 Report

Role of dorsomedical hypothalamus GABAergic neurons in sleep-wake states in response to changes in ambient temperature in mice

Lei Li et al.

This study investigates into the duration of sleep-wake states when the animals are exposed to cold or hot environments. Core body temperature is one of the essential biological rhythms that display the metabolic changes that are induced by wake or sleep states. Because dorsomedial hypothatlmus is involved in the thermoregulation, it is safe to assume that the wake-sleep states would be affected by changes in the activity. The authors report that DMH is indeed a regulator of the core body temperature and that the inhibitory neurons play an important role in the sleep-wake changes.

1. The time window of the exposure to different ambient temperatures varies between the low ranges and the high ranges. The authors conducted the cold exposure experiments during the light phase, when the animals are most sleepy, and exposed the animals to warm exposure during the dark phase when the animals are more active. Is there any reason why the experiments were conducted such way?

2. Did you see any changes in the delta power and their decay rate? Were there any differences in the transitional states? From wake to sleep and vice versa?

3. Were there any metabolic changes after the acute exposure? Especially after the cold exposure?

4. In the introduction, the authors briefly mentioned the effects on the circadian rhythm. Could you explain how this could be involved with acute changes to the ambient temperatures in the discussion?

Reviewer 2 Report

In order to elucidate the important role of dorsomedial hypothalamus (DMH) in the regulation of sleep related to ambient temperature, the authors firstly examined the effects of cold/warm exposure on core temperature and sleep-wake cycle, and secondly assessed the effects of DMH-lesion (caspase-3-induced apoptosis or chemogenetics) on core temperature and sleep-wake cycle under cold/warm exposure. The present study concluded that DMH GABAergic neurons have an important role in the regulation of sleep-wake behaviors elicited by a change in ambient temperature.

The findings obtained by this study are considerably of interest. However, several points need clarifying. These are given below.

Major comments:

  1. In page 18, lines 442 to 458, the authors described the possible reasons why the present findings are not consistent with the previous data obtained by Thomas et al. (2003, ref.#37). But, I guess that the inconsistency is strongly due to the differences in housing condition, especially light/dark condition. Thomas et al. (2003) examined the effects of ibotenic acid-induced lesion of DMH neurons on sleep/wake cycle under constant darkness, in order to aviod the confounding effect of environmental light on biological rhythms. So that, it is important to emphasize that this difference could result in the discrepancy between the present data and the previous findings. Additionally, the authors should mention that the reason why the animals were housed under 12/12h light/dark condition in the present study, despite of the massive influence of environmental light on biological rhythms in mice.

  1. In connection with comment #1, why this study did cold/warm exposure exclusively conduct at resting/active period, respectively? I understand that the cold exposure at light period enhances wake, but the warm exposure at dark period enhances NREM sleep from Figure 1 and 2. However, it is considerably of interest whether cold/warm exposure at dark/light affects sleep/wake cycles. This point should be discussed.

  1. The authors judged that the apoptosis-induced DMH lesions did not change the rhythms of core temperature and sleep/wake cycle. Although the sleep/wake parameters in DMH-lesioned mouse were not statistically significant compared to control, it seems that the DMH-lesioned mouse became drowsy/sleepy at dark period (Figure 4L, N and Figure 6L, N). This may be due to the large variance. Above-mentioned reports described by Thomas et al. (2003) measured the degree of damage in DMH-neurons and found that the changes in biological rhythms of core temperature and sleep/wake cycle related to the loss of DMH-neurons. So that, in the present study, the large variance in sleep/wake parameters in DMH-lesioned mouse is due to the individual differences in the number of remaining DMH neurons. I guess that additional data and analysis for mentioning should improve the quality of this manuscript.

  1. In the present study, the chemogenetic inactivation of DMH-neurons resulted in the decrease in REM sleep during 12h light period. I would like the authors to explanate why this change occurs from the standpoint of the relations between sleep/wake cycle and variable subpopulations of DMH neurons.

  1. In page 17 to 18, lines 430 to 431, the authors desribed that “The inactivation of these neurons an increase NREM sleep during the inactive period”. But, I wonder what data indicated it. From the Figure 8P, the NREM sleep was not changed by chemogenetic inactivation of DMH-neurons.

  1. In page 13, lines 331 to 333, the authors described that c-Fos expression was induced by the CNO injection. Because this data was not presented, please add it.

  1. In page 2, lines 86 to 88, the authors described that “at 12°C, wakefulness increased slightly compared with that at 16°C (Fig.1E-G)”. However, I guess that the wakefulness at 12°C was slightly decreased compared with that at 16°C (Fig.1E). Please check and correct it as needed.

  1. In this text, a lot of mismatches between the descrition of results and correspondent Figure are found. For example, in page 9, lines 204 to 207, “(Fig. 4B-D)” is not suitable for the expression of this data, but “(Fig.3K and 3N)” is correct. The authors must check carefully and revise properly.

Minor comments:

1) in page 3, line 104, “gradient cold exposure”-> “gradient warm exposure”.

2) in page 7, lines 164 to 165, and in page 10, lines 227 to 228, “Lower panels” -> “Left pannels”, and “upper panels” -> “right pannels”.

3) in page 7, line 167, and in page 11, lines 230 to 231, “Scale bars in the upper panel…” was not correct.

4) in page 7, lines 169 to 170, and in page 11, lines 232 to 233, “(indicated as filled bars)” and “(indicated as open bars)” were upside down.

5) in page 8, line 181 and 187, “cold exposure” -> “warm exposure”.

6) in page 13, line 331, the authors described “CNO (500 nM)”. But in page 14, line 343 and in Figure 7E, “CNO (5 µm)” was noted. Please correct it.

7) in page 17, line 381, “hM3Dq” -> “hM4Di”, “the activation” -> “the inactivation”, and “increased” -> “decreased”.

8) in page 17, lines 387 to 389, please add “** p < 0.01”.

9) in page 17, line 392, “DMH-vGAD-hM3Dq” -> “DMH-vGAD-hM4Di”.

10) in page 18, line 470, and in page 19, line 526, “°C” was double.

11) There were many unneeded hyphen in the “Materials and Methods” section. Please correct them (in page 18, line 472; in page 19, lines 487, 491, 496 and 516; in page 20, lines 535, 544, 547, 559, 560, 563, 567 and 568; inpage 21, lines 582, 592, 594 and 597).

12) in page 21, line 582, I wonder what “optical stimulation” indicated. The present study did not conduct optogenetic intervention.

13) in page 21, lines 602 to 603, I wonder what “except for nest-building test” indicated. The present study did not conduct nest-building test.

14) the descriptions of reference list were confusing. Please correct them according to the submission guidelines.

I hope these comments will be helpful.

Round 2

Reviewer 1 Report

The manuscript seems to be improved greatly, however I still found some pieces puzzling.   1. It is more clear now; however please show in the supplementary results of the effects from the different schedules of changes In ambient temperatures.   2. The power densities show some slight differences; however it is hard to see the differences due to the way it is displayed in the figure.  

Reviewer 2 Report

The authors clearly responded to my comments and suggestions. Lastly, I suggest a minor revision, "(Figure 6B, D)" -> "(Figure 6B-E)" in page 14, line 4.
